# Differentiation of High-Level Language Semantics

**Michael Innes**
Julia Computing, Inc.
Edinburgh, UK
mike.j.innes@gmail.com

## Abstract

Though analytic differentiation (AD) is a program transformation, AD tools have typically supported only very limited program representations, consisting of primitive mathematical operations and basic structured control flow. Zygote, an AD for the Julia language, instead operates on Julia code. This presents an interesting challenge for the AD implementor: the program representation now contains not just mathematical operations, but arbitrary control flow, user-defined functions, recursion, data structures, mutation, metaprogramming, foreign function calls, specialised hardware, and even concurrency and parallelism primitives. This paper explains how Zygote handles these high-level features safely and efficiently, making an unusually large set of Julia programs differentiable.

## 1 Introduction

Reverse-mode analytic differentiation (AD) is a program transformation [18]; AD tools transform an input program (the primal) into a new program that calculates derivatives (the adjoint)[1]. However, AD tools have typically only supported simple program representations, consisting only of variable assignments and mathematical operations (in AD terminology, a 'Wengert list', [21]; in machine learning a 'graph' or 'trace'). More complex programming languages, such as Python and C++, are supported by *tracing* operations in the original language into a simpler representation; recent machine learning frameworks also augment the trace with basic structured control flow.[2]

Zygote [10] achieves performance and flexibility by differentiating Julia's syntax tree (AST) ahead of time. However, this means that it must handle any language feature that could appear in a differentiable program, from closures to concurrency. Zygote's core is a program transformation that handles arbitrary control flow, function calls and user-defined gradients; this paper details how we build on this foundation to support everything else.

Notable related work includes Tapenade [5] and Swift for TensorFlow [20], which support a useful subset of Fortran and Swift's semantics respectively, alongside Stalin∇ [16] and Myia [19], which extend AD with support for closures and recursion.

## 2 Custom Adjoints

Zygote's core transform is simple and mechanical, and only around 200 lines of code. Almost all of Zygote's semantics and functionality are provided via its library of custom adjoints, which have somewhat surprising expressive power. Defining a custom adjoint is similar to defining a normal

---

[1]In 'dynamic' or 'eager-mode' ADs, like autograd [14], the program transformation is more implicit, since it is interleaved with numerical evaluation. The adjoint trace need not be fully realised except when nesting derivatives.

[2]A paradigmatic example of this format is the XLA intermediate representation, [1].

33rd Conference on Neural Information Processing Systems (NeurIPS 2019), Vancouver, Canada.

Julia function, except that the definition must return both the output of the function and a *pullback* closure, which propagates gradients as in [11].

```
@adjoint a * b = (a * b, dc -> (dc'b, dc'a))
```

Alongside mathematical gradients, we can define utilities such as gradient hooks, which allow an arbitrary function to be applied to the gradient. For example, `hook(-, x)` reverses the sign of $\bar{x}$; more generally it can also be used for gradient clipping and debugging.

```
hook(f, x) = x
@adjoint hook(f, x) = (x, dx -> (nothing, f(dx)))
```

The function `nestlevel` is able to do reflection on the gradient process itself; if called within a differentiated function it will return the order of differentiation being performed.

```
nestlevel() = 0
@adjoint nestlevel() = (nestlevel()+1, _ -> nothing)
```

A simple implementation of checkpointing is similarly straightforward.

```
checkpoint(f, x) = f(x)
@adjoint checkpoint(f, x) = (f(x), dy -> J(f, x)[2](dy))
```

All functionality in this paper is similarly implemented via custom adjoints.

## 3 Data Structures & Mutation

The most fundamental data structure is the *cons cell*, a tuple of two values like $C = (x_1, x_2)$. If we call `first(C)` to retrieve the first element we must then find the gradient with respect to $C$ in the adjoint program. We create an *adjoint object* $\bar{C}$, which mirrors the structure of $C$ while storing the gradient of each internal element $(\bar{x}_1, \bar{x}_2)$, so the adjoint for $y = \text{first}(C)$ is $\text{cons}(\bar{y}, 0)$. This naturally generalises across different numbers of fields or names of accessor functions.

To handle mutation, consider a one-element 'box' structure $B$. We can $\text{get}(B)$ to retrieve the current stored value, and $\text{set}(B, x)$ to erase that value and replace it with $x$. The adjoint object $\bar{B}$ is also a box. The pullback for `get` accumulates the gradient in the adjoint box while the pullback for `set` returns it, resetting the adjoint to $0$. Any data structure can then be modelled as a combination of cons cells and boxes, though more efficient or direct representations and adjoints can easily be provided. Since the adjoints for data structures do not capture their inputs by value, later mutations do not invalidate the pullback.

Closures are just objects with a `call` method [4]; the fields of the object represent the closure's environment. In our compiler all functions actually accept a hidden environment argument—which may be empty as a special case—so both closures and higher-order functions are supported with no extra effort.

## 4 Concurrency and Parallelism

Julia supports a concurrency model based on communicating sequential processes (CSP, [6]). A zero-argument function or closure (a thunk) can be scheduled as a *task* (or *coroutine*), and executed independently of the main thread. Tasks communicate with each other through shared queues called *channels*. Typically, the main thread will create a series of tasks and wait for them all to finish before continuing.

Zygote makes CSP differentiable by the following transformation. Firstly, when a task is scheduled, its thunk $f$ is replaced by $\mathcal{J}(f)$, producing a pullback. Once the task is complete, we associate it with an adjoint task which will run the pullback. During the reverse pass, we reach the point where the original task was awaited in the primal code, and schedule the adjoint task. The adjoint task executes and communicates with other adjoint tasks as needed, finally producing a gradient of the thunk $\bar{f}$. Channels can be differentiated as in §3; for each channel $c$ we create an empty adjoint channel $\bar{c}$. Sending a value to $c$ becomes receiving a sensitivity from $\bar{c}$ and vice versa.

Julia supports shared-memory parallelism by multiplexing tasks onto OS threads, so support for tasks means that multithreaded code is also differentiable. Julia uses the same concepts, though a slightly different API, for distributed / multi-node parellelism, so the same techniques can be straightforwardly transferred to differentiation of distributed code. In an experimental setting we were able to achieve a $1.5\times$ speedup when using two cores to get the gradient of a simple function using map-reduce parallelism.

Care must be taken that write/accumulate operations in the adjoint are atomic, since there may otherwise be a race condition due to multiple reads from the same array location in the primal. Differentiation of parallel code at other levels of abstraction, such as the level of parallel `for` loops or map-reduce, presents different challenges and opportunities [9, 8, 15, 3].

# 5 Mixed-Mode AD

Alternatives to reverse-mode AD have advantages in many situations, even when calculating gradients. For example, Julia's forward-mode AD [17] has constant memory overhead (compared to reverse mode's tape, linear in the number of instructions executed) and has minimal time overhead, making it ideal for long-running computations with a small number of inputs. Similarly, TaylorSeries.jl [2] can calculate arbitrary-order forward-mode derivatives in one shot. Mixed mode is exposed by writing `forwarddiff(f, x)`. This calculates the same result as $f(x)$, but additionally calculates the Jacobian via forward mode, stores it, and applies it during the backwards pass using a custom adjoint. Similarly, checkpointed AD is exposed via `checkpoint(f, x)` (§2). Zygote can be instructed to always use forward mode (or another AD technique) on a given function, or even to have heuristics for the best method, so that for users of a library, differentiation is efficient by default.

An equivalent problem is differentiating code in other languages, for example Python code invoked via PyCall.jl [13]. In this case, we can write an adjoint for the low-level `pycall` function which invokes a Python AD, capturing its tape in a pullback. To a user, calling imported Python functions inside a call to `gradient` then works transparently.

# 6 Complex Differentiation

Zygote defines the sensitivity of a complex number $z = x + yi$ by $\bar{z} = \bar{x} + \bar{y}i$. This definition is useful for gradient descent since for small, real $\eta$, $f(z + \eta\bar{z}) \approx f(z) + \eta\bar{z}\bar{z}^*$, and thus the usual gradient update $z := z - \eta\bar{z}$ lowers the loss. (This is equivalent to differentiating a pair of two reals $(x, y)$.) Zygote's pre-defined rules for numerical operations (like $\times$ and $+$) are automatically consistent with this definition, so a only rule for one of `real`, `imag` or conj is needed in addition for full complex support.

This sensitivity is not the true complex derivative $\frac{\partial}{\partial z} = \frac{\partial}{\partial x} + \frac{\partial}{\partial iy} = \frac{\partial}{\partial x} - i\frac{\partial}{\partial y}$, which (for holomorphic functions) will satisfy $f(z + \epsilon) \approx f(z) + \frac{\partial f}{\partial z}\epsilon$. By the Cauchy-Riemann equations, $\frac{\partial f}{\partial z}$ is conjugate to the sensitivity $\bar{z}$ of $\Re f(z)$ making it straightforward and efficient to calculate. In the more general non-holomorphic case one needs either the equivalent $2 \times 2$ real Jacobian or the two Wirtinger derivatives $(\frac{\partial f}{\partial z}, \frac{\partial f}{\partial z^*})$, both of which are readily derived from the sensitivities of $\Re f(z)$ and $\Im f(z)$.

# 7 Staged Programming

Zygote works well with Julia's excellent meta-programming facilities. For example, many numerical libraries provide an `einsum` interface, allowing tensor operations to be expressed with a syntax based on Einstein notation. The syntax is usually expressed as a string and, in dynamic interfaces like PyTorch, parsing the string incurs an overhead each time the expression is run. In Julia, Einsum can be implemented as a *macro*, explicitly parsing the notation at compile time and leaving only raw tensor operations behind. Zygote sees only the final matrix multiply and sum operations, so this has no overhead compared to writing them manually. The same is true of Julia's other powerful metaprogramming and staging tools, such as generated functions.

## 8 Hardware Backends

Zygote transforms generic programs and mathematical expressions – written in terms of mathematical operators like $\times$, $+$ etc. – into new generic programs that calculate a gradient. Thus Zygote is completely agnostic to the data types running through the program and how they are implemented or represented in memory. A Zygote program written for floating point numbers therefore works equally well with rational numbers, arbitrary-precision floats and integers, measurements, hardware-specific types like `BFloat16`, and combinations of these.

```julia
julia> gradient(x -> x^2 + 3x + 1, 1/3)
(3.6666666666666665,)

julia> gradient(x -> x^2 + 3x + 1, 1//3)
(11//3,)

julia> gradient(x -> x^2 + 3x + 1, 1/3 ± 0.01)
(3.6666666666666665 ± 0.02,)
```

The same is true for arrays; the program `gradient(x -> σ.(W*x .+ b), x)` works equally well whether $W$, $b$ and $x$ are dense arrays, sparse arrays, arrays backed by GPU memory, or distributed arrays stored over a cluster of hundreds of nodes. The operations $\times$, broadcasting and so on are called on the adjoint arrays and thus launched on the GPU or cluster as appropriate.

## 9 External Libraries

Support for types and libraries distinguishes frameworks from programming languages, and we support these in differentiable programming too. For example, the Colors.jl package [7] provides representations of RGB colours (among many other colour spaces), and functions over these colour spaces can be differentiated, even when they comprise hundreds of lines of code and many language features.

```julia
julia> a = RGB(1, 0, 0); b = RGB(0, 1, 0);

julia> gradient(a -> a.r^2, a)
((r = 2.0f0, g = nothing, b = nothing),)

julia> colordiff(a, b)
86.60823557376344

julia> gradient(b -> colordiff(a, b), b)
((r = -1.77, g = 28.88, b = -0.04),)
```

Aside from the correctness benefits of working with types, it is increasingly recognised that incorporating existing knowledge and code into machine learning leads to richer and more powerful models; this is particularly valuable in scientific computing, where powerful explicit models exist for many systems that need not be learned from scratch [12].

## 10 Conclusion

We have demonstrated how Zygote, an AD for the Julia language, can support features of a high-level programming language. We also show how the same techniques can be used to add advanced features to Zygote, such as checkpointed, mixed-mode and cross-language AD, without changes to Zygote's core source transform. We believe that Zygote's unusual extensibility makes it an appealing target for research into advanced AD techniques.

Languages for differentiable and probabilistic programming need not sacrifice modern programming language design, or be restricted to simple DSLs. We hope that this work can inform the design both of ADs for existing languages, and of new languages and IRs designed to support machine learning from the ground up.

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
