# OpenReview forum: "Differentiation of High-Level Language Semantics"
_NeurIPS.cc/2019/Workshop/Program_Transformations — Program Transformations @NeurIPS2019 Poster_

### Official Review · AnonReviewer2 · 2019-09-26
**Impressive set of supported language constructs in AD**

**Confidence:** 5
**Rating:** 6

**Review:**

Although the introduction does mention some other examples (like Tapenade and Swift for TensorFlow) I still think that the first section slightly misrepresents the state of AD: Performing AD by transforming the source code (or some representation thereof) has been around for a long time and has been implemented on a variety of languages such as Fortran, Java, Matlab, etc. (see https://arxiv.org/abs/1502.05767 for an overview). In that sense, Zygote and other recent frameworks such as Swift for TensorFlow are probably best presented as new implementations of old and tested techniques (that maybe had fallen out of favor in ML).

That said, while I see little novelty in the underpinnings of the framework, the range of supported features is impressive and suggests that the implementation details of the framework were thought through and well-executed. As such, I would love to see this work presented in more detail at the workshop. I would suggest that the authors focus the presentation of their work on advanced features such as concurrency/parallelism, checkpointing, and mixed-mode AD. I would also like to see a discussion of the frameworks shortcomings or unsupported features, e.g., how does Zygote handle mutability of arrays or dynamic redefinition of functions?

---

### Official Review · AnonReviewer1 · 2019-09-30
**Solid implementation of AD in Julia**

**Confidence:** 5
**Rating:** 7

**Review:**

The article is a good overview of all the AD features that have been made to work in Zygote, an implementation of AD in Julia. The whole language (including recursion and higher-level functions), excluding macros themselves, is supported.
Additional features have also been implemented, such as checkpointing, hooks, and mixed-mode AD.
It is interesting to see that simple design decisions on the representations of data (cells and boxes) and derivatives (pullback closures) enabled different extensions.
It would be great to learn about how those concepts needed to be specialized or optimized for different use cases, such as sparse arrays, asynchronous execution.

---

### Public Comment · ~Andreas_Griewank2 · 2019-10-02
**There was serious AD long before ML**

Dear Michael Innes,
          Please do not be so brash while reinventing the wheel. Believe me we in AD have been talking forever about "Graphs and Traces" long before ML existed, almost never about "Wengert Lists", which is supposedly "our" AD terminology. Also we have of course dealt with dynamic control flow, recursion, user defined functions, parallelism, mixed mode differentiation etc.
          Moreover, we have worried about what the numerical values coming out of a formally differentiated program mean, if anything. Especially at the cracks or in the case of iterative solvers. That does not seem to concern you at all, even though Julia is supposed to have super numerics.
          Your disregard for previous work does not betray any kind of professional courtesy.
           Andreas Griewank
P.S. I published a paper on optimal checkpointing in 1991.

---

> ### Public Comment · ~Michael_Innes1 · 2019-10-08
> **Agreed**
>
> Dear Andreas,
>
> Sorry that the paper came across that way; it was not the intention.
>
> I absolutely agree that AD work done outside of ML – including much of your own – builds the foundation for everything we do. We don't think any of these features are novel in themselves, or even that they haven't been done much better before (as is certainly the case for things like checkpointing and iterative solvers). Our goal was only to present a particular set of design decisions that we considered interesting, in a particular SCT AD implementation.
>
> I hope you'll be happier with the longer (in progress) version of the paper, which discusses the relationship to existing work in AD much more thoroughly, and I'd be very happy to take feedback and suggestions on it. Feel free to reach out any time if you're interested in that.
>
> Thanks,
> Mike

---

### Decision · Program_Chairs · 2019-10-01

**Decision:**

Accept (Poster)

**Comment:**

An overview of a strong AD system for Julia, but lacking significant novelty.